# β-Hydroxybutyrate Increases Exercise Capacity Associated with Changes in Mitochondrial Function in Skeletal Muscle

**DOI:** 10.3390/nu12071930

**Published:** 2020-06-29

**Authors:** Matías Monsalves-Alvarez, Pablo Esteban Morales, Mauricio Castro-Sepulveda, Carlos Sepulveda, Juan Manuel Rodriguez, Mario Chiong, Verónica Eisner, Sergio Lavandero, Rodrigo Troncoso

**Affiliations:** 1Laboratorio de Investigación en Nutrición y Actividad Física (LABINAF), Instituto de Nutrición y Tecnología de los Alimentos (INTA), Universidad de Chile, Santiago 8380492, Chile; matias.monsalves@inta.uchile.cl (M.M.-A.); csepulvedag@ug.uchile.cl (C.S.); juan.rodriguez@inta.uchile.cl (J.M.R.); 2Advanced Center for Chronic Diseases (ACCDiS), Facultad de Ciencias Químicas y Farmacéuticas & Facultad de Medicina, Universidad de Chile, Santiago 8380492, Chile; pmc.bqk@gmail.com (P.E.M.); mchiong@ciq.uchile.cl (M.C.); 3Escuela de Kinesiologia, Facultad de Medicina, Universidad Finis Terrae, Santiago 7500000, Chile; m.castro.med@gmail.com; 4Departamento de Biología Celular y Molecular, Facultad de Ciencias Biológicas, Pontificia Universidad Católica de Chile, Santiago 8330025, Chile; veisner@bio.puc.cl; 5Corporación Centro de Estudios Científicos de las Enfermedades Crónicas (CECEC), Santiago 7860201, Chile; 6Department of Internal Medicine (Cardiology Division), University of Texas Southwestern Medical Center, Dallas, TX 75390-8573, USA

**Keywords:** ketone bodies, β-hydroxybutyrate, mitochondrial morphology, skeletal muscle, endurance

## Abstract

β-hydroxybutyrate is the main ketone body generated by the liver under starvation. Under these conditions, it can sustain ATP levels by its oxidation in mitochondria. As mitochondria can modify its shape and function under different nutritional challenges, we study the chronic effects of β-hydroxybutyrate supplementation on mitochondrial morphology and function, and its relation to exercise capacity. Male C57BL/6 mice were supplemented with β-hydroxybutyrate mineral salt (3.2%) or control (CT, NaCl/KCl) for six weeks and submitted to a weekly exercise performance test. We found an increase in distance, maximal speed, and time to exhaustion at two weeks of supplementation. Fatty acid metabolism and OXPHOS subunit proteins declined at two weeks in soleus but not in tibialis anterior muscles. Oxygen consumption rate on permeabilized fibers indicated a decrease in the presence of pyruvate in the short-term treatment. Both the tibialis anterior and soleus showed decreased levels of Mitofusin 2, while electron microscopy assessment revealed a significant reduction in mitochondrial cristae shape in the tibialis anterior, while a reduction in the mitochondrial number was observed only in soleus. These results suggest that short, but not long-term, β-hydroxybutyrate supplementation increases exercise capacity, associated with modifications in mitochondrial morphology and function in mouse skeletal muscle.

## 1. Introduction

Ketone bodies (KB) are lipid-derived molecules produced by the liver under starvation as an alternative energy substrate for the brain, heart, and skeletal muscle (SkM) [1]. Initially, KB were thought to be harmful metabolic wastes [2,3], but nowadays KB and especially β-hydroxybutyrate (β-HB) are gaining attention due to their therapeutic potential [4]. β-HB has been shown to reduce oxidative stress [5,6,7], inflammation [8,9,10], and protein degradation [11], while improving cardiac efficiency by increasing myocardial blood flow and heart rate in humans [12,13,14]. Besides, the athletic community has recently taken a particular interest in β-HB, as it has been proposed as a new ergogenic aid [15]. The addition of β-HB in the form of mono-, di-esters, or salts (i.e., bound to Ca^2+^, Na^+^, K^+^) to a standard or high carbohydrate diet may confer an improvement in performance without the adverse effects of carbohydrate restriction [16,17]. In rodents, the chronic supplementation of β-HB has a positive impact on their lipid profile [18,19], weight reduction [20,21], as well as an enhancement in running capacity [22], but the mechanisms are unknown. In humans, Cox et al., showed that the acute ingestion of a ketone ester (d-β-hydroxybutyrate-r, 1, 3-butanediol monoester) improved endurance capacity by 2% in highly trained cyclists [23]. Interestingly, athletes receiving the supplementation showed glycogen preservation and an increase in intramyocellular lipid utilization after an endurance time trial test, suggesting the role of acute nutritional ketosis in metabolism preference during exercise [24]. On the other hand, acute β-HB supplementation impairs high-intensity performance despite an increase in fatty acid oxidation [25], suggesting that the type of exercise must be considered when the supplementation is considered.

β-HB oxidation is performed in the mitochondrial matrix of extrahepatic organs. Among those tissues, SkM plays an important role as the main site for KB utilization during resting conditions [3]. Interestingly, different SkM types possess different metabolic activity for KB. For instance, oxidative muscles with high mitochondrial content, such as the soleus (SOL), have higher ketolytic protein content than glycolytic fibers. However, their expression can be induced by chronic endurance exercise training [26,27]. The increase in KB uptake during exercise is related to a concomitant higher mitochondrial enzymatic capacity, evidenced by citrate synthase activity. The latter may explain why trained mice show a high β-HB oxidation rate compared to sedentary controls [28]. However, little is known about changes in substrate utilization and their relationship to mitochondrial bioenergetics by the chronic exogenous intake of exogenous KB on skeletal muscle.

Mitochondria are known to form a highly dynamic and interconnected network in adult SkM [29]. Mitochondria can adjust their morphology upon different energetic challenges induced by a nutritional stimulus, such as nutrient excess or deprivation [30]. A fragmented mitochondrial network is observed with an excess of caloric intake [31]. However, under starvation, an elongated mitochondrial network is induced [32]. Mitochondrial fission is regulated mainly by the recruitment of the GTPase dynamin-related protein Drp1 to the outer mitochondrial membrane (OMM), that allows for mitochondrial constriction and later separation [33]. Mitochondrial fusion is regulated in a two-step process: The first is governed by mitofusins Mfn1 and Mfn2, which are responsible for the fusion of the OMM. The second step is controlled by optic atrophy gene 1 (Opa1), that promotes inner mitochondrial membrane (IMM) fusion [34,35,36]. The fusion of mitochondria is more active in oxidative SkM fibers, while it is less frequent in glycolytic SkM and is enhanced by oxidative substrates [37]. Mitochondrial fusion and fission protein levels seem to be modified by diet composition [38], suggesting that mitochondrial morphology relies on nutrient supply and the energy demands of each muscle type. Furthermore, mitochondrial morphology is intimately related to their function [39]. Therefore, this work aims to study the effects of sustained β-HB supplementation in endurance capacity in mice, focusing on mitochondrial morphology and function.

## 2. Materials and Methods

### 2.1. Chronic β-HB Supplementation

Animal care and procedures were approved by the Animal Ethics Committee of the University of Chile. Male C57BL/6 mice were obtained from the Institute of Public Health (Santiago, Chile) at three weeks of age. Mice were housed at four animals/cage for the first three weeks and then placed in individual cages for one week before the beginning of the short- (two weeks) and long-term (six weeks) supplementation experiments. They were acclimatized under a 12:12 light/dark cycle and kept between 21 and 24 °C. Mice were fed ad libitum during the whole study with a standard diet (LabDiet: 5P00, Prolab 3000), which contained 59% kcal from carbohydrates, 14% kcal from fat, and 26% kcal from protein. After six weeks, the mice were randomly assigned to a control (CON) group or β-HB supplementation (β-HB) group. The β-HB group was supplemented with a commercially available β-HB Na^+^/K^+^ (racemic mixture of D- and L-β-HB) solution (Ketoforce, Ketosport, USA). This product has been used recently to stimulate hyperketonemia in healthy subjects [25,40]. The dosage for available β-HB supplementation was based on previous literature, according to the manufacturer [18,19]. The β-HB solution contained 39% *w*/*v* of β-HB, 5.3% *w*/*v* sodium and 5.3% *w*/*v* potassium, which was diluted in the drinking water to obtain a final concentration of 3.2% *w*/*v* of β-HB, 0.45% *w*/*v* sodium and 0.45% potassium. CON drinking water was supplemented with a NaCl and KCl solution to match the concentration for both cations to that of the drinking water from the β-HB group. Both solutions were available throughout the experimental period and were changed every four days.

### 2.2. Exercise Performance Test

All mice were familiarized to run on a motorized treadmill without the electric grind. For this purpose, a ball was placed in the rear of each lane to encourage running with reduced stress [41]. Treadmill familiarization was performed three days in a row for 10 min with a velocity of 10 m/min. For the endurance test, mice were placed in the room for 5 min to allow for environmental acclimation and located on their lanes. The treadmill exercise started at 5 m/min with an inclination of 5° during the whole test. Speed was increased by 1.7 m/min every 12 min until exhaustion, which was determined as the inability to maintain the speed despite the rear ball stimulation [42]. Each mouse was returned to its cage after exhaustion, where food and water were available. This procedure was repeated every seven days to record weekly changes in each mouse. Endurance performance was determined by determining the maximal distance (m), time to exhaustion (min), maximal speed (m/min), and vertical work (kg·m). The latter was measured as the product of vertical distance and mouse body weight, as previously described [43]. Food intake was assessed by feeding a weekly amount of food per week (40 g), and pellets and food remnants on the cage were weighed daily on a precision scale. Animal euthanasia was performed 48 h after the last endurance test, the mice were euthanized with isoflurane (Forane^©^, Abbott), followed by cervical dislocation. The tibialis anterior (TA) and SOL muscles were extracted and frozen in liquid nitrogen.

### 2.3. Grip Strength Test

Total strength was determined by an all-grip strength test measured in a force gauge (Force Gauge, China). Mice were placed horizontally over the metal grid and pulled by the tail continuously until they could not hold on anymore. This procedure was repeated three times for each mouse to obtain a mean value. Strength determination was done one day before endurance test days and performed in all mice once a week during the whole study. Grip strength was expressed in N (Newtons) and normalized by body weight (N/g) [42].

### 2.4. Blood Ketone Measurements

Blood collection was performed between 7:00–8:00 a.m. by a small cut on the tip of the tail [44] after an overnight fast. Blood β-HB levels were determined once a week using a Freestyle NEO monitor system (Abbott, IL, USA). All measurements were performed following the manufacturer’s instructions.

### 2.5. Transmission Electron Microscopy

Skeletal muscle samples were prepared as described previously [35]. After mice were euthanized, TA and SOL were removed and immediately fixed in 2.5% glutaraldehyde for 2 h at room temperature, followed by overnight incubation at 4 °C. The next day, muscles were dissected into small bundles of fibers, washed four times with 0.1 M sodium cacodylate buffer, and stained with 2% osmium tetroxide in 0.1 M sodium cacodylate buffer for 2 h. Samples were then washed with water and stained with 1% uranyl acetate for 2 h. Stained samples were dehydrated in an acetone dilution series and embedded in Epon. Then, 80 nm thin sections were cut from the embedded specimens, mounted on electron microscopy grids, and examined using a transmission electron microscope (Philips, Tecnai 12) at 80 kV. Morphometric analysis of intermyofibrillar mitochondria (IMF) was performed on longitudinal sections using Fiji/ImageJ software. The quantification of number, size, and density of mitochondria was assessed at 8200X magnification, while mitochondrial ultrastructure was visualized at 20,500X magnification. The classification of crista morphology was determined as indicated in previous work [45], and mitochondrial density was determined by the ratio between mitochondrial and tissue, as described by del Campo et al. [46]. 

### 2.6. Skeletal Muscle Fiber Permeabilization (PmFB)

Mitochondrial respiratory parameters were studied in situ in fresh saponin-permeabilized fibers, as described previously [47]. Briefly, TA and SOL muscles were extracted and placed in ice-cold BIOPS buffer containing (in mM): K_2_EGTA 7.23, CaK_2_EGTA 2.77, imidazole 20, DTT 0.5, taurine 20, ATP 5.7, phospho-creatine 14.3, MgC1_2_ 6.56, and MES 50 and adjusted to pH 7.1 and kept at 25 °C. Following separation, the bundles were placed in BIOPS buffer plus saponin (50 μg/mL) and incubated at 4 °C for 30 min. Fibers were then washed twice with Z-buffer containing K-MES 105 mM, KCl 30 mM, EGTA 1 mM, KH_2_PO_4_ 10 mM, MgCl_2_ 5 mM, 5 mg/mL fatty acid-free BSA, blebbistatin 20 μM, and creatine 20 mM, adjusted to pH 7.4 and kept at 4 °C until the respiration assay.

### 2.7. Mitochondrial Oxygen Consumption

After permeabilization, the oxygen consumption rate was measured using a Clark electrode (Oxygraph+, Hansatech Instruments, King′s Lynn, UK). Then, 1–3 mg (wet weight) of muscle fiber bundles were placed inside the oxygen chamber containing Z-buffer at 25 °C. After 2 min of stabilization, substrates/inhibitors were added to the chamber in sequential order: pyruvate (5 mM) plus malate (2 mM), ADP (1 mM), glutamate (5 mM), succinate (5 mM) and rotenone (complex I inhibitor, 10 μM). For fatty acid oxidation palmitoyl–carnitine was used (0.02 mM) + malate (2 mM), and ADP (1 mM) [47]. Oxygen consumption rates were expressed as nmol of O_2_/mg wet weight, as previously described [47].

### 2.8. Western Blot Analysis

Frozen SkM was thawed and immediately homogenized in RIPA lysis buffer (Tris-HCl 10 mM, pH 7.4; EDTA 5mM; NaCl 50 mM; sodium deoxycholate 0.1%; triton X-100 1% *v*/*v*) supplemented with protease and phosphatase inhibitors (Roche Applied Sciences, Germany) using a motorized pestle system (Pellet Pestle Cordless Drive Unit, DKw Life Sciences, USA). The total lysate was centrifuged at 12,000 rpm three times for 10 min at 4 °C; the supernatant was collected for protein determination using BCA assay (Novagen, Merck Ma, NJ, USA). Protein separation and immunoblotting were performed, as previously described [42]. Immunoblotting was performed using the following antibodies: Mfn2 (ab50838, Abcam, 1:1000), Mfn1 (ab126575, Abcam, 1:1000), Opa1 (ab42634, Abcam, 1:1000), Drp1 (PA1-16987, Thermo, 1:1000), phosphoserine 616-Drp (5616, Cell Signaling, 1:1000), Total OXPHOS cocktail (ab110413, Abcam, 1:1000), Mic60 (#10179-1-AP, Proteintech, 1:1000), Mic19 (#25625-1-AP, Proteintech, 1:1000), mHsp70 (#MA3-028, Pierce, 1:1000), Oxct1 (#12175-1-AP, Proteintech, 1:1000), Acca2 (#100847, Santa Cruz, 1:1000), Scad (#365953, Santa Cruz, 1:1000), Vlcad (#376239, Santa Cruz, 1:1000), Cpt1 (#393070, Santa Cruz 1:1000), Tom20 (72610, Cell Signaling, 1:1000), and β-Tubulin (T0198, Sigma Chemical Co, 1:5000). Samples of the same muscle were run on the same gels.

### 2.9. RNA Isolation and RT-PCR

Muscle tissue total RNA was isolated using the TRIzol^®^ method. The concentration of the extracted RNA was determined spectrophotometrically, and 1 μg RNA was used for each reverse transcriptase reaction. cDNA was prepared with OligodT primers and a RevertAid First Strand cDNA Synthesis Kit (Thermo Scientific, Waltham, MA, USA). Real-time qPCR reaction was performed using the KAPA SYBR FAST kit (Roche Diagnostics, Basel, Switzerland), according to the manufacturer’s instructions in an ECO-Illumina Real-Time PCR System (Roche Diagnostics, Basel, Switzerland). The primers′ sequences are presented in Appendix A. Data were analyzed as described previously [48] and are expressed as fold change in gene expression in comparison to the control. All of the experiments were repeated four times, and the geometric mean of β-Actin and YWHAZ were used for normalization [49]. 

### 2.10. Statistical Analysis

All procedures were performed with 5–14 animals, depending on the experimental setting. Data are presented as mean ± standard error of the mean (SEM). Statistical significance was determined by unpaired Student’s *t*-test or two-way analysis of variance (ANOVA), followed by Bonferroni′s post hoc. Statistical significance was set at *p* < 0.05. All analyses were performed using GraphPad Prism software (version 7.0, San Diego, CA, USA).

## 3. Results

### 3.1. β-HB Supplementation Improves Endurance Capacity

Six weeks of β-HB supplementation did not affect body weight compared to the control group at the time of sacrifice (CON 24 ± 0.76 g and β-HB 24 ± 0.46 g). Blood β-HB levels were not different at baseline but were higher at two and six weeks of experimentation between groups (*p* = 0.001 and *p* = 0.037, respectively), as shown in Table 1. The β-HB group showed a decrease in weekly food intake at the second week of treatment (CON 2.7 ± 0.1 g/d; β-HB 2.2 ± 0.1 g/d, *p* = 0.020), while calorie intake did not differ between groups (*p* = 0.25). There was no difference in the amount of drinking water consumed by each group (CON 8.8 ± 1.3 mL/d; β-HB 9.0 ± 1.1 mL/d, *p* = 0.171). Since the supplement was added to drinking water, we estimated the mean daily β-HB intake to be 320.4 ± 54.4 mg/d, which includes 0.60 kcal that were considered and added to the total caloric intake for the supplemented mice, presented in Table 1.

To investigate whether β-HB affects endurance performance, we performed an acute treadmill test every seven days at the end of each week over a six-week experimental period. Running distance was no different at baseline but was significantly greater in the second week (*p* = 0.033) on β-HB, when compared to CON treated mice. β-HB treated mice showed an improvement from baseline to the second week (*p* < 0.0001) but a loss at the end of the treatment, as shown in Figure 1A. Regarding the time to exhaustion, mice receiving β-HB ran 28.4% more (*p* = 0.049), as shown in Figure 1B, showed higher vertical work (*p* = 0.026), as shown in Figure 1C, maximal speed (*p* = 0.025), as shown in Figure 1D, and relative strength (*p* = 0.017, not shown) than CON mice, only in the second week.

### 3.2. β-HB Long-Term Supplementation Altered Electron Transport Chain Subunit Protein Content

TA and SOL SkM fibers differ in morphological and metabolic characteristics, the former being more glycolytic and displaying lower mitochondrial density, while SOL presents a higher amount of mitochondrial and oxidative potential [50,51]. Based on those differences, the protein content of the electron transport chain (ETC) subunits on TA, shown in Figure 2A,B, and SOL, shown in Figure 2H,I, were assessed after six weeks of β-HB supplementation. The β-HB-treated group showed a significant increase in SDHB (complex II subunit) and ATP5A (ATP synthase subunit) protein content compared to CON in the glycolytic TA muscle, as shown in Figure 2B, with no changes in the mitochondrial protein mHsp70, used as a marker of mitochondrial mass, as shown in Figure 2C [52,53]. Changes in the oxidative SOL muscle were only observed in ATP5A, as shown in Figure 2I, with a non-statistical difference in mHsp70 protein levels, as shown in Figure 2J.

### 3.3. β-HB Long-Term Supplementation Induces a Mild Change in Mitochondrial Dynamics Proteins

As SkM mitochondria use KB for energy production, we sought to determine if long-term β-HB supplementation was able to change the content of mitochondrial dynamics and ketolytic proteins. We found no changes in Mfn2, shown in Figure 2E, or Opa1, shown in Figure 2F, levels in TA compared to CON, although we did find a reduction in the mitochondrial fission protein Drp1, as shown in Figure 2G. On the contrary, β-HB treatment in SOL muscle did not alter Mfn2 or Drp1 protein levels, as shown in Figure 2L,N, but significantly increased Opa1 levels, as shown in Figure 2M.

### 3.4. β-HB Short-Term Supplementation Modifies Mitochondrial Function and Substrate Utilization 

Since we observed an increase in running capacity in the β-HB-supplemented mice after two weeks of treatment, as shown in Figure 1, we wanted to determine if these changes are related to modifications in mitochondrial function in response to short-term exposure to β-HB. Using SkM permeabilized fiber bundles (PmFB), we assessed the differences in O_2_ consumption and substrate oxidation after short-term β-HB intake, as shown in Figure 3. TA showed a significant decrease in pyruvate-sustained respiration, as shown in Figure 3A, and no changes in the presence of succinate, as shown in Figure 3B, or succinate plus complex I rotenone inhibitor, as shown in Figure 3C, or with palmitoyl-carnitine, as shown in Figure 3D, compared to CON. Interestingly, no changes in OXPHOS subunit protein content were observed in TA, as shown in Figure 3F. Similar to TA, the SOL SkM fibers presented a significant decrease in pyruvate oxidation, as shown in Figure 3G, and no changes in succinate or succinate plus rotenone, as shown in Figure 3H,I. No change in oxygen consumption was observed in the presence of palmitoyl–carnitine oxidation in SOL in the β-HB supplemented group compared to CON, as shown in Figure 3J. In contrast to TA, SOL OXPHOS subunits CI, CII, and CIII presented a significant decrease, compared to the non-supplemented group, as shown in Figure 3L.

### 3.5. Lipid Metabolism and Ketone Oxidation Protein Content Are Altered After Short-Term β-HB Supplementation

SkM phenotypes possess different compositions and, thereby, metabolic preferences [51]. Based on the observed changes in PmFB respiratory measurements, as shown in Figure 3, we assessed the protein and gene expression of relevant fatty acid and ketone utilization enzymes, shown in Appendix A. TA showed no changes in FA mitochondrial transport as determined by carnitine palmitoyl transferase 1b (Cpt1b), as shown in Figure 4A, compared to CON. Levels of very-long-chain acyl-coenzyme a dehydrogenase (Vlcad), the enzyme responsible for the first step of β-oxidation, were significantly reduced, as shown in Figure 4B. In contrast, short-chain-acyl CoA dehydrogenase (Scad), as shown in Figure 4C, or Acetyl-Coenzyme A acyltransferase 2 (Acaa2), as shown in Figure 4D, levels were not altered. As observed in TA, SOL muscle showed no changes in Cpt1b, as shown in Figure 4G, or Acaa2, as shown in Figure 4J, but we found a significant decrease in Vlcad, as shown in Figure 4H, and Scad, as shown in Figure 4I, protein levels. Interestingly, both TA, as shown in Figure 4E, and SOL, as shown in Figure 4K, showed an increase in Scot protein levels, suggesting than ketone supplementation alters SkM oxidative metabolism.

### 3.6. β-HB Short-Term Supplementation Alters Mitochondrial Dynamic Fusion Protein Content

Based on the small changes we observed in mitochondrial fission and fusion proteins at the end of the six-week treatment, as shown in Figure 2, we determined if the peak of performance time observed after two weeks of β-HB supplementation was related to changes in these proteins. In TA muscle, as shown in Figure 5, Mfn2 levels decreased significantly compared to CON, as shown in Figure 5B, while Mfn1, as shown in Figure 5C, and Opa1 levels remained unchanged, as shown in Figure 5D. Mitochondrial fission protein Drp1 also showed no change in phosphorylation status measured by pDrp1^Ser616^, as shown in Figure 5H. Furthermore, no differences were observed in mitochondrial mass assessed by mHsp70, as shown in Figure 5E, or TOM20, as shown in Figure 5I. In the SOL muscle of β-HB-fed mice, as shown in Figure 5J, Mfn2 and Opa1, as shown in Figure 5K,M, significantly decreased compared to CON, and no changes on Mfn1 were observed, as shown in Figure 5L. Interestingly, mHsp70 protein levels were lower in SOL from β-HB-fed mice, as shown in Figure 5N, while TOM20 showed a mild increase, as shown in Figure 5R.

In terms of fission, the levels of pDrp1^Ser616^ showed no significant change (*p* = 0.0601), as shown in Figure 5Q. As cristae shape is fundamental to sustain oxidative phosphorylation [54], we determined the level of protein components of the mitochondrial contact site and crista organizing system (MICOS) that play a critical role in IMM bending and crista junction formation. In TA muscles from β-HB-fed mice, MIC19 but not MIC60 showed differences compared to CON, as shown in Figure 5G,F. Interestingly, SOL showed that MIC60, but not MIC19, increased significantly, as shown in Figure 5O,P, implying a muscle-type differential mitochondrial response to daily β-HB intake.

### 3.7. β-HB Short-Term Supplementation Disrupts Mitochondrial Morphology of SkM with Different Phenotypes

The mitochondrial shape is altered by different metabolic stressors [30]. As we were able to observe a change in mitochondrial proteins in response to β-HB treatment, the determination of mitochondrial content and shape could be useful to explain the endurance response induced by β-HB supplementation. Transmission Electron Microscopy (TEM) was performed on TA, as shown in Figure 6A, and SOL, as shown in Figure 6F, after two weeks of β-HB supplementation. Under this condition, TA presents changes in mitochondrial size, as shown in Figure 6B, but no alterations in density, as shown in Figure 6C, or number, as shown in Figure 6D. Notably, significant changes were observed in crista shape in the β-HB group, as the number of abnormal cristae (low electron-density and cristae continuity) compared to CON was increased, as shown in Figure 6E. SOL muscle exhibited no change in mitochondrial size, as shown in Figure 6G, or density, as shown in Figure 6H, while we found a mild increase in abnormal cristae, as shown in Figure 6J, and a decrease in mitochondrial number, as shown in Figure 6I.

## 4. Discussion

This work provides new evidence of the short- and long-term effects of β-HB exogenous supplementation, with a special focus on their impact on skeletal muscle mitochondrial morphology and function. Here we showed that β-HB supplementation can induce an improvement in exercise capacity and grip strength and that these enhancements are associated with changes in mitochondrial morphology and function.

### 4.1. Sustained Supplementation of β-HB Increases Physical Performance without Affecting Body Weight

Our results showed that β-HB supplementation had no impact on body weight or caloric intake during the whole experimental protocol, despite transiently reducing food intake in week two, which was corrected by the calorie intake of the supplement. The chronic intracerebroventricular infusion of β-HB has shown to reduce weight and food intake in Sprague–Dawley rats [55]. Regarding ketone supplementation, ketone ester and β-HB salt have reported a similar impact in reducing body weight [19,56], but studies have not reported on the food intake patterns of the animals. Similarly, Caminhotto et al., recently showed a trend to decline in the food intake, accompanied by a reduction in adipocyte volume after four weeks of β-HB salt supplementation, without an alteration of body weight [18]. We did not measure adipocyte volume or distribution. Still, there was no difference in epididymal fat weight between conditions at the end of the short- or long-term protocols, which suggests no significant changes in overall adiposity (data not shown).

As β-HB in our study was supplemented in drinking water and not by gavage, as in others [19,57,58], we cannot discount the possibility that the sensory properties of the water could affect food intake in the treated group. Nevertheless, blood β-HB levels increased significantly in the experimental group, suggesting that mice were, in fact, consuming the supplement during the whole study. It is important to mention that the β-HB supplement used in our experiments consisted of 50% d-β-HB and 50% l-β-HB [59], the former being highly metabolizable, while l-β-HB metabolism is still poorly studied [60,61,62]. Importantly, plasma β-HB values obtained in our study are similar to those reported with ketogenic diets in mouse models [63,64], indicating that this protocol is an effective way to induce mild hyperketonemia without a reduction in or restriction of carbohydrates. Thus, our supplementation protocol proved to be useful to elucidate some of the effects induced by β-HB on endurance capacity.

We observed an improvement in time to exhaustion, maximal speed, vertical work, and total distance in short (two weeks) but not in long-term (six weeks) β-HB supplementation, suggesting that the elevation of β-HB induces metabolic changes in a more reduced, rather than prolonged time, which could have implications for the practical timeframe of supplement utilization. These results are in line with those presented by Murray et al., who also reported a transient increase in running distance in rats fed a β-HB-rich diet followed by a decline [22]. This decay might be attributed to slow recovery time between the trials, as opposed to our model, tested every seven days. Additionally, Murray et al. showed that β-HB increased workload and the free energy for ATP hydrolysis in the heart, providing a possible mechanism to explain why non-trained rats augment their exercise capacity chronically, similar to what we observed in our model.

### 4.2. Short-Term β-HB Supplementation Reduces Pyruvate Oxidation

Exogenously ingested β-HB has been shown to acutely increase fatty acid oxidation in humans [25], mainly because β-HB can be oxidized to acetoacetate (AcAc), elevating acetyl-CoA and citrate levels and by inhibiting pyruvate dehydrogenase (PDH) and phosphofructokinase (PFK), reducing glycolytic flux and allowing lipid metabolism to increase [65].

With the PmFB methodology, we were able to observe metabolic changes depending on SkM phenotypes. SOL and TA permeabilized fiber showed a significant decrease in oxygen consumption with pyruvate as a substrate, suggesting a reduction in its oxidation. In the presence of palmitoyl–carnitine, only TA tended to increase, while SOL muscle showed the opposite response. A limitation of the present study is the absence of the β-HB incorporation on the PmFB protocol to test the changes in oxygen consumption rate associated with its oxidation after the supplementation. However, a similar protocol using EDL muscle fibers, using AcAc as a substrate, showed an increase in the oxygen consumption rate, changes in mitochondrial morphology, and membrane potential [37].

Mice that received β-HB in our study were submitted to an incremental 12-minute stage running protocol, allowed for the usage of lipids and ketones as primary fuel sources [66,67]. Accordingly, the elevation of β-HB through KD has been shown to increase acute exercise endurance capacity in C67BL/6 mice. These changes were associated with modifications in ketolytic and lipolytic enzymes, such as Bdh1, Oxct1, and Atgl and Hsl mRNA levels, which were differentially expressed between SkM types [67]. Our results suggest that short-term β-HB intake alters β-oxidation enzyme protein levels, especially on highly oxidative muscle, such as SOL, and to a lesser degree in the glycolytic TA. We found a decrease in long-chain fatty acid Vlcad protein on both muscles, while Scad protein levels decayed only on SOL. Vlcad and Scad (Acyl-CoA dehydrogenases) deficiencies are related to diminished fatty acid oxidation [68,69]. The decrease in both proteins could explain why SOL presents a lesser response to palmitoyl–carnitine oxygen consumption, while TA seems to increase in our model.

Regarding FA metabolism gene expression, we only observed a decrease in ACSL mRNA levels on TA, while no other changes in FA transport, lysis or Lcad, and Hadha were detected, as shown in Appendix A. However, we cannot discard the fact that sustained β-HB salt ingestion could lead to unwanted side effects on fatty acid metabolism, as we only measured these parameters in the two-week treatment. Notably, since β-HB oxidation does not rely on β-oxidation to yield acetyl-CoA for the TCA cycle, the increase in Scot observed on both SkM suggests than the main fuel source was β-HB. Further investigations regarding the chronic consumption of β-HB and the metabolic differences between their forms (i.e., Mono, di-esters) and in vivo oxidation will help to elucidate the specific consequences on lipid metabolism.

### 4.3. β-HB Supplementation Modifies Mitochondrial Morphology

Exercise and nutrition can alter cellular metabolism through the modification of mitochondrial dynamics [30,70]. Until now, the mitochondrial response in the presence of β-HB has not been explored in a model without energy restriction. We observed that mitochondrial mass, assessed by mHsp70 and TOM20 levels, was decreased after two weeks of β-HB supplementation only in SOL muscle, which was recovered after six weeks. The changes observed in the ETC protein levels at six weeks post-treatment are more remarkable in TA 6 than in SOL, in which the changes were observed after two weeks. Interestingly, β-HB intake modulates the TCA cycle and ETC intermediate levels when it is metabolized with or without energy restriction [71]. The reaction mediated by Scot during ketolysis results in the donation of CoA from succinyl-CoA by the TCA cycle flux [72]. This could explain why mice were able to run longer despite SOL producing a decrease in OXPHOS protein levels. As Scot protein levels increased in both types of SkM, the increase in ketone body degradation may enhance the generation of the intermediate and maintain mitochondrial activity or improve energy production during starvation or carbohydrate restriction.

Regarding mitochondrial fusion proteins, the most remarkable change was at two weeks, when Mfn2 levels were significantly decreased in both glycolytic and oxidative SkM. This suggests that acute treatment with this metabolite stresses mitochondria, possibly leading to an alteration in mitochondrial dynamics and shape in SkM. Notably, altered mitochondrial morphology has been associated with improved endurance capacity [73].

Regarding mitochondrial morphology, we showed here that two weeks of β-HB intake altered cristae shape and decreased mitochondrial number. In contrast, mitochondrial size increased in TA with more subtle changes in SOL. Increased mitochondrial size seems inconsistent with lower levels of Mfn2. However, mitochondrial size also relies on volume regulation, which may be altered in this case, considering the disorganization of mitochondrial cristae. It has been shown that both the increase in and decay of fatty acid oxidation can alter mitochondrial metabolism and morphology, mainly by acute stress induced by mitochondrial ROS [74,75]. Our data suggest that exogenous β-HB ingestion leads to a myriad of metabolic-related modifications (i.e., oxygen consumption in response to pyruvate or PC, β-oxidation enzyme gene expression, Scot protein levels) in a SkM-specific manner. This implies that our TEM results on mitochondrial morphology alteration could be in part associated with substrate preference modifications, especially through ketone metabolism.

We acknowledge some limitations to our study. For instance, a more detailed exploration of mitochondrial activity in response to β-HB as substrate, and skeletal muscle ketone oxidation during exercise, could help discern whether the increase in performance endurance reported here is a consequence of the direct effect of β-HB on mitochondrial physiology or a result of upstream metabolic modifications. Additionally, the evaluation of markers of mitophagy and protein turnover on SkM with different forms of exogenous ketones will aid in integrating the metabolic and signaling responses elicited by β-HB.

## 5. Conclusions

In summary, we provide new information about the effects of β-HB intake on mitochondrial morphology and function, stressing the differences among SkM phenotypes. Our results showed that glycolytic phenotypes, such as those of the tibialis anterior, could be more sensitive to KB metabolism, inducing a change in their metabolism. This leads to alterations in mitochondrial shape that meet energetic challenge demands, such as those imposed by exercise. Given the importance of mitochondrial ability to support whole-body metabolism and the increase in the popularity of exogenous ketones and KD, a more detailed understanding of the effect of different forms of β-HB on skeletal muscle is relevant to understand exercise performance and different skeletal muscle-related diseases.

## Figures and Tables

**Figure 1 nutrients-12-01930-f001:**
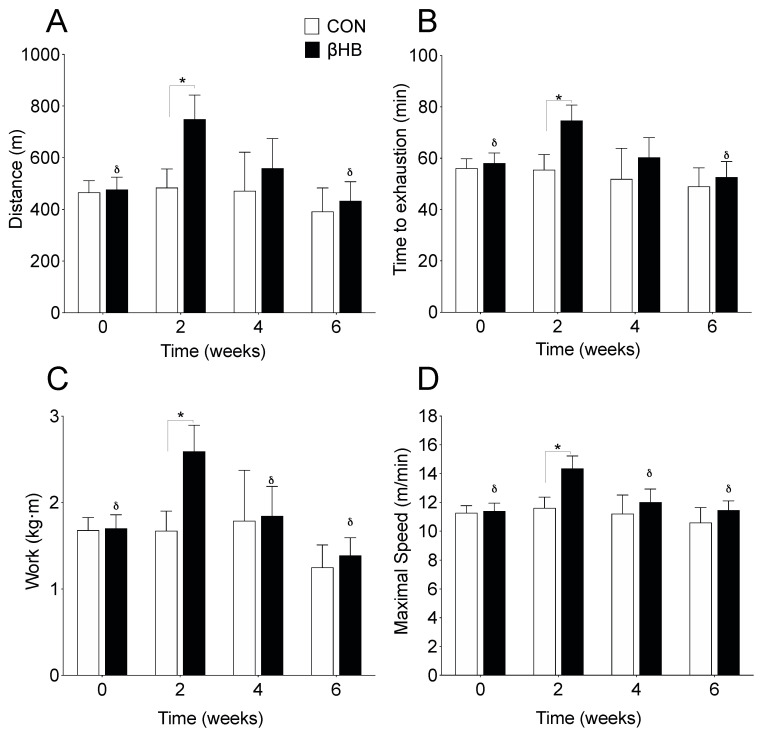
Effect of β-hydroxybutyrate supplementation on physical performance. (**A**) Running distance. (**B**) Time to exhaustion. (**C**) Vertical work. (**D**) Maximal speed. *n* = 8–14 mice per condition. * *p* < 0.05 between CON and β-HB; δ, *p* < 0.05 compared to the second week within the β-HB group. Two-way ANOVA repeated measurements with Bonferroni′s post hoc test. Values expressed as mean ± SEM.

**Figure 2 nutrients-12-01930-f002:**
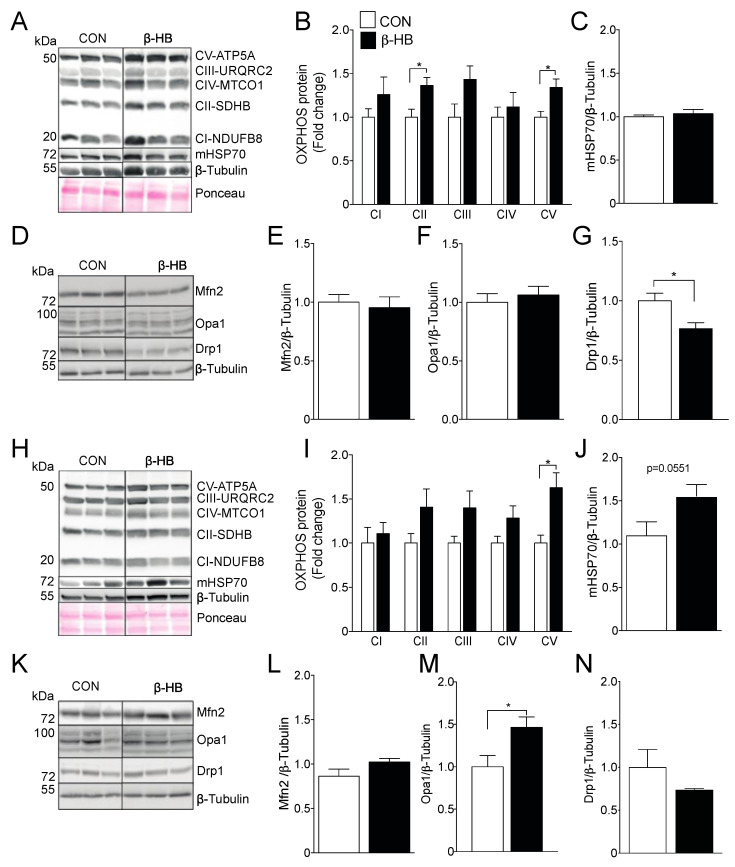
Six weeks of β-HB intake alter the electron transport chain and mitochondrial dynamics protein levels. (**A**) Representative OXPHOS Western blot from Tibialis anterior whole tissue extract. (**B**) Tibialis Anterior OXPHOS quantification. (**C**) mHsp70 levels. (**D**) Tibialis anterior mitochondrial fission and fusion proteins representative Western blot. (**E**) Mfn2. (**F**) Opa1 and (**G**) Drp1. (**H**) Soleus representative OXPHOS Western blot. (**I**) Soleus OXPHOS quantification. (**J**) mHsp70. (**K**) Soleus mitochondrial fission and fusion proteins representative Western blot. (**L**) Mfn2. (**M**) Opa1 and (**N**) Drp1. The grouping of blots cropped from different parts of the same gel were divided by black lines. *n* = 5–6 mice per condition, * *p* < 0.05, Unpaired *t*-Test. Values expressed as mean ± SEM.

**Figure 3 nutrients-12-01930-f003:**
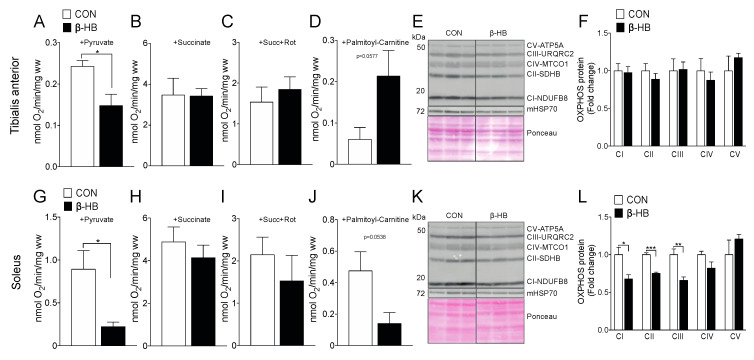
On PmFB, β-hydroxybutyrate alters oxygen consumption by reducing pyruvate oxidation, but modifies palmitoyl–carnitine utilization depending on SkM. Tibialis anterior PmFB oxygen consumption in the presence of: (**A**) pyruvate, (**B**) pyruvate oxidation + succinate, (**C**) pyruvate oxidation + succinate + rotenone, (**D**) palmitoyl–carnitine oxidation. (**E**) Tibialis anterior representative OXPHOS Western blot. (**F**) Tibialis anterior OXPHOS subunit quantification. (**G**) Soleus PmFB oxygen consumption in the presence of pyruvate (**G**), (**H**) pyruvate oxidation + succinate, (**I**) pyruvate oxidation + succinate + rotenone, (**J**) palmitoyl–carnitine oxidation. (**K**) Soleus representative OXPHOS Western blot; (**L**) Tibialis anterior OXPHOS subunit quantification. The grouping of blots cropped from different parts of the same gel were divided by black lines. *n* = 3–5, * *p* < 0.05, ** *p* < 0.01, *** *p* < 0.001, Unpaired *t*-Test. Values expressed as mean ± SEM.

**Figure 4 nutrients-12-01930-f004:**
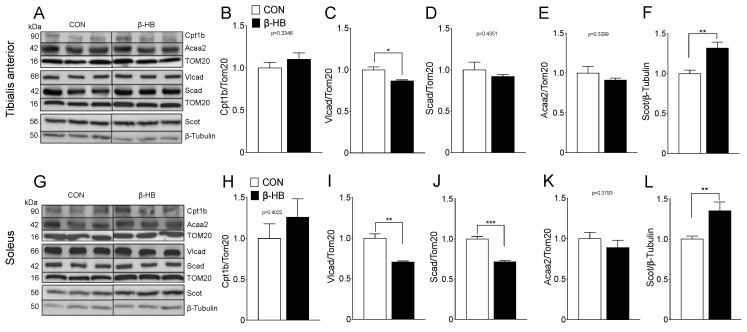
β-Oxidation and β-hydroxybutyrate oxidation protein content are altered mainly on SOL after a two-week β-hydroxybutyrate intake. Bar graphs represent mean band intensity. (**A**) Representative Western blot for tibialis anterior. (**B**) Cpt1b. (**C**) Vlcad. (**D**) Scad. (**E**) Acca2. (**F**) β-HB utilization rate-limiting enzyme Scot. (**G**) Representative Western blot for Soleus. (**H**) Cpt1b. (**I**) Vlcad. (**J**) Scad. (**K**) Acca2. (**L**) β-HB utilization rate-limiting enzyme Scot. The grouping of blots cropped from different parts of the same gel were divided by black lines. *n* = 4–6 mice per condition, * *p* < 0.05, ** *p* < 0.01, *** *p* < 0.001, Unpaired *t*-Test. Values expressed as mean ± SEM.

**Figure 5 nutrients-12-01930-f005:**
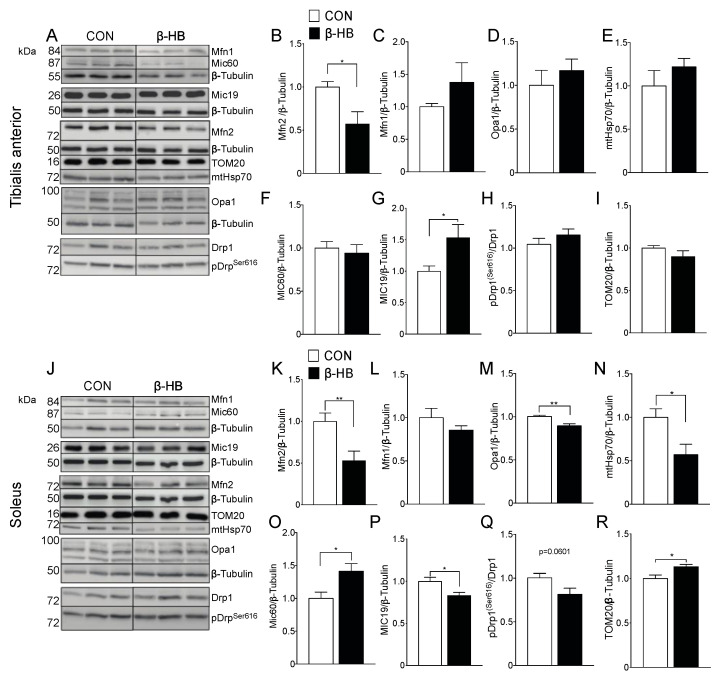
Short-term β-hydroxybutyrate intake modulates fission and fission protein content differentially among SkM phenotypes. (**A**) Tibialis anterior whole tissue extract. Representative Western blot for fusion proteins. Bar graphs represent mean band intensity. (**B**) Mfn2. (**C**) Mfn1. (**D**) Opa1. (**E**) mHsp70. (**F**) Mic60. (**G**) Mic19. (**H**) pDrp^ser616^. (**I**) TOM20. (**J**) Soleus representative Western blot. Bar graphs represent mean band intensity. (**K**) Mfn2. (**L**) Mfn1. (**M**) Opa1. (**N**) mHsp70. (**O**) Mic60. (**P**) Mic19. (**Q**) pDrp^ser616^. (**R**) TOM20. The grouping of blots cropped from different parts of the same gel were divided by black lines. *n* = 4–10 mice per condition, * *p* < 0.05, ** *p* < 0.01, Unpaired *t*-Test. Values expressed as mean ± SEM.

**Figure 6 nutrients-12-01930-f006:**
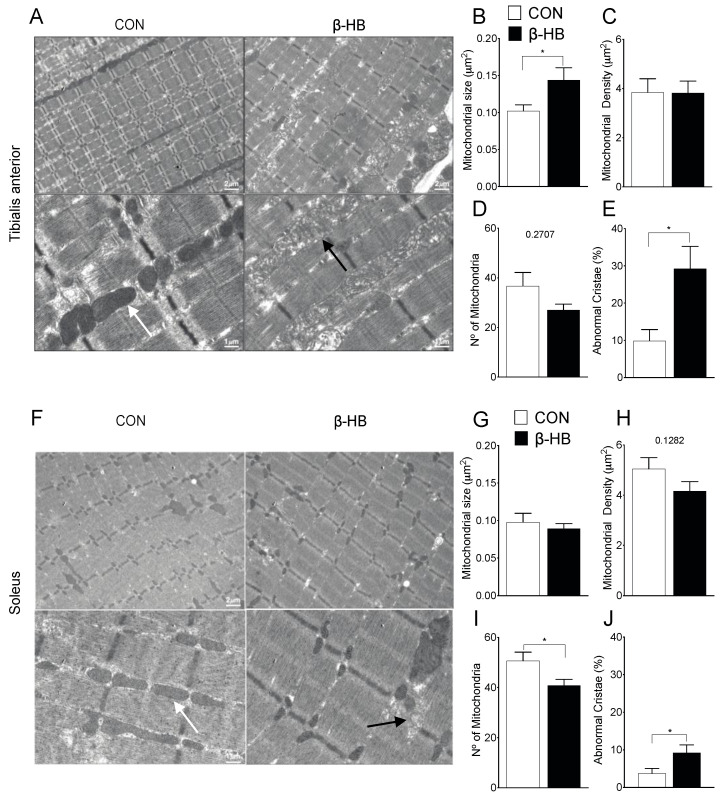
Short-term β-hydroxybutyrate diet alters mitochondrial morphology. **Tibialis anterior:** (**A**) Representative electron microscopy of both groups. (**B**) Mitochondrial size μm^2^. (**C**) Mitochondrial density (%). (**D**) Mitochondria number. (**E**) Abnormal cristae. **Soleus:** (**F**) Representative electron microscopy of both groups. (**G**) Mitochondrial size μm^2^. (**H**) Mitochondrial density (%). (**I**) Mitochondria number. (**J**) Abnormal cristae. * *p* < 0.05, Unpaired *t*-Test. Values expressed as mean ± SEM. >15 mitochondria per fiber, >4 animals per condition. Scale bars: 1 and 2 μm, as indicated. The white arrows indicate normal and the black arrows abnormal mitochondria.

**Table 1 nutrients-12-01930-t001:** β-hydroxybutyrate supplementation effects in body weight, blood β-hydroxybutyrate and caloric and food intake.

Parameter		CON	β-HB	*p*-Value
Body weight (g)	Basal	22.5 ± 0.48	23.8 ± 0.40	0.073
	2 weeks	23 ± 0.61	23 ± 0.25	0.987
	4 weeks	24 ± 0.60	24 ± 0.36	0.788
	6 weeks	24 ± 0.76	24 ± 0.46	0.945
β-hydroxybutyrate (mM)	Basal	0.4 ± 0.8	0.4 ± 0.04	0.765
	2 weeks	0.4 ± 0.04	0.7 ± 0.03	0.001 *
	4 weeks	0.4 ± 0.04	0.7 ± 0.08	0.056
	6 weeks	0.5 ± 0.06	0.8 ± 0.09	0.037 *
Calorie intake (kcal)	2 weeks	10.6 ± 0.41	9.2 ± 0.62	0.120
	4 weeks	11.1 ± 0.54	10.5 ± 0.75	0.532
	6 weeks	11.2 ± 0.21	11.6 ± 0.29	0.379
Food intake (g)	2 weeks	2.7 ± 0.10	2.2 ± 0.15	0.020 *
	4 weeks	2.8 ± 0.14	2.5 ± 0.19	0.266
	6 weeks	2.9 ± 0.05	2.9 ± 0.06	0.832

Control group (CON), β-hydroxybutyrate supplemented group (β-HB). Values expressed as mean ± SEM. * *p* < 0.05.

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
