# Peer review of "β-Hydroxybutyrate Increases Exercise Capacity Associated with Changes in Mitochondrial Function in Skeletal Muscle"

_nutrients, 2020, doi:10.3390/nu12071930_

Round 1
Reviewer 1 Report
The authors have studied to find whether b-HB increases exercise-capacity and mitochondrial function in skeletal muscle and stated that b-HB-induced alteration of mitochondrial function improves running endurance. Below are some general and specific comments the authors may wish to consider.
- Introduction;
The authors properly described their goals in the introduction.
- Method
- Although the authors provided the composition of solution that contained b-HB but should provide how much they consumed daily.
- The authors are required to consistently fill out the country label for purchased items, for example, KAPA SYBR FAST kit (Roche Diagnostics, Switzerland).
- Result
- Line 204, the authors describe glycemia, but I can’t find glycemia data.
- In Figure 1, the running endurance capacity by b-HB appears to be significantly lower than the basal level. The authors should state this in the result section.
- The authors should define all the factors when this shows up the first time, for example, mitochondrial heat shock protein 70 (mtHSP70), which plays a role in the control of cell proliferation.
- The authors presented all the blot in figures that were cut between con and b-HB. The authors should provide all the whole (intact) blot for reviewers to check.
- The authors should use mHSP70 or mtHSP70 in figures and text.
- Line 268, “-HB” must be “β-HB”.
- Since the authors used the word “short-term” or “long-term”, should define as “short-term (2 weeks) and long-term (6 weeks)” at the method section.
- Use the same font and size in figures 4 and 5.
- Discussion
- The improvement of running distance capacity induced by b-HB was gradually lowered depend on time (Fig. 1), I think it was affected by the b-HB-induced disruption of mitochondrial morphology (Fig. 7I and 7J) since the running endurance capacity clearly links to type I muscle function. Running distance in b-HB at 4 weeks was almost the same as the basal level. I don’t think running capacity was improved. And I also think resting for 6 days for mice is enough to recover. The Authors should discuss this in more detail.
- Though I could not find the glycemia data that is lowered by b-HB, is this mean that b-HB decreased lipolysis in adipose tissue? How muscle can use fatty acid in this situation? In figure 4, b-HB decreased OXPHOS enzymes and function in soleus (Fig. 4G~4L).ETC is unlikely to be able to make enough ATP for muscle contraction, although muscle possibly accelerates the TCA cycle by feed acetyl CoA using b-HB, the author should illustrate in more detail how the mice ran better.
- Line 428 and 432, please check this word. “Vldcad”
- Line 452, “OMM and IMM membrane”, remove the membrane.
Author Response
Manuscript ID: Nutrients-833855-R1
Manuscript title: β-hydroxybutyrate increases exercise capacity associated with changes in mitochondrial function in skeletal muscle
Reviewer ‘comments:
Query regarding language: English language and style are fine/minor spell check required
Reply: An English native speaker edited the MS.
REVIEWER 1
The authors have studied to find whether b-HB increases exercise-capacity and mitochondrial function in skeletal muscle and stated that b-HB-induced alteration of mitochondrial function improves running endurance. Below are some general and specific comments the authors may wish to consider.
- Introduction
The authors properly described their goals in the introduction.
- Method
Query 1: Although the authors provided the composition of solution that contained b-HB but should provide how much they consumed daily.
Reply: We corrected the average daily intake of mice in the ?-HB group (refer to lines 208-209).
Query 2: The authors are required to consistently fill out the country label for purchased items, for example, KAPA SYBR FAST kit (Roche Diagnostics, Switzerland).
Reply: As suggested, we added the corresponding country of every purchased item.
- Result
Query 3: Line 204, the authors describe glycemia, but I can’t find glycemia data.
Reply: We corrected this mistake. We assessed glycemia in the short-term (2-week) experiments. However, as we modified one of the figures, we find this data irrelevant to the current version of the manuscript.
Query 4: In Figure 1, the running endurance capacity by b-HB appears to be significantly lower than the basal level. The authors should state this in the result section.
Reply: We appreciate this particular query regarding endurance capacity. We found no statistical difference at baseline on time or distance (p>0.99) between groups. This point is described now in the results section (see lines 213-218). We also added a new set of mice to improve the statistical power (see New Fig. 1).
Query 5: The authors should define all the factors when this shows up the first time, for example, mitochondrial heat shock protein 70 (mtHSP70), which plays a role in the control of cell proliferation.
Reply: We used mitochondrial Hsp70 (mHsp70) (along with Tom20) as a mitochondrial mass housekeeping protein, as suggested by others for skeletal muscle tissue (DOI: 10.1016/j.exger.2014.08.013, DOI:10.1152/ajpendo.00043.2007, DOI: 10.1152/ajpcell.00181.2010). We added this rationale in lines 231-232
Query 6: The authors presented all the blot in figures that were cut between con and b-HB. The authors should provide all the whole (intact) blot for reviewers to check.
Reply: We understand this concern. We included a file with the whole (intact) membranes of each experiment, along with a description in Methods (lines 182-183)
Query 7: The authors should use mHSP70 or mtHSP70 in figures and text. Line 268, “-HB” must be “β-HB”.
Reply: Both corrections have been made in the final manuscript, as suggested.
Query 8: Since the authors used the word “short-term” or “long-term”, should define as “short-term (2 weeks) and long-term (6 weeks)” at the method section.
Reply: We changed the description of experimental times in the methods section (see line 92).
Query 9: Use the same font and size in figures 4 and 5.
Reply: As suggested, we have modified this typo error.
- Discussion
Query 10: The improvement of running distance capacity induced by b-HB was gradually lowered depend on time (Fig. 1), I think it was affected by the b-HB-induced disruption of mitochondrial morphology (Fig. 7I and 7J) since the running endurance capacity clearly links to type I muscle function. Running distance in b-HB at 4 weeks was almost the same as the basal level. I don’t think running capacity was improved. And I also think resting for 6 days for mice is enough to recover. The Authors should discuss this in more detail.
Reply: We appreciate this particular query regarding the performance results in our manuscript. We have responded partially to this concern in your Query 4. As more mice were added to the experiments, we observed a more significant effect on the performance-enhancing properties of β-HB, specifically in mice supplemented for a short timd (2 weeks). We agree with your statement, “The improvement of running distance capacity induced by b-HB was gradually lowered depending on the time I think it was affected by the b-HB-induced disruption of mitochondrial morphology (Fig. 7I and 7J)”, as our results showed this might be linked to mitochondrial morphology. A robust change in diet composition might represent a stressing condition for mitochondria. Our work shows both a fiber type and temporal specific mitochondrial response, and it remains puzzling why mitochondrial cristae ultrastructure shows a derangement in a significant number of organelles, as well as augmented size after two weeks of b-HB diet in the Tibialis Anterior muscle.
Nevertheless, bioenergetic performance was positively adapted in a short term diet as well as in a long-term diet. However, exercise performance was not significantly improved in the more prolonged diet (4th or 6th week) as opposed to the short-term (2nd week). One possible explanation is that with the short-term diet, mitochondria adapt comparably as do hyper-fused mitochondria upon ROS stimulation (DOI: 10.1038/emboj.2009.89) or cigarette smoke (DOI: 10.1093/toxsci/kfw102), that involve increased oxidative metabolism. Although little is known about the adaptive mechanisms undertaken by mitochondria to overcome challenging environmental changes, long term adaptations allow mitochondria to endure these changes (DOI: 10.1038/s41556-018-0133-0).
Query 11: Though I could not find the glycemia data that is lowered by b-HB, is this mean that b-HB decreased lipolysis in adipose tissue? How muscle can use fatty acid in this situation? In figure 4, b-HB decreased OXPHOS enzymes and function in soleus (Fig. 4G~4L).ETC is unlikely to be able to make enough ATP for muscle contraction, although muscle possibly accelerates the TCA cycle by feed acetyl CoA using b-HB, the author should illustrate in more detail how the mice ran better.
Reply: We appreciate the comment. In Query 4 we described that glycemia was measured in the short term (2 weeks) protocol, and is not included in the current manuscript.
We do not assess adipose tissue lipolysis in our study. However, b-HB is a nicotinic acid receptor inhibitor, reducing fatty tissue triglyceride hydrolysis when ketones are elevated (DOI:10.1074/jbc.C500213200). We measured epididymal fat after the supplementation period with no changes among groups (CON 0.36±0.07g, b-HB0.37±0.5, p=0.692) (see lines 359-361).
As you mention, soleus from b-HB treated mice showed a decrease in OXPHOS protein levels in short-term (2 weeks) treatment (Fig.3L). However, there were no changes in Tibialis Anterior Fig.3F). We proposed that Scot (increased in both SOL and TA, Fig.4F and Fig.4L, respectively) may play a key role in sustaining TCA intermediates by ketone oxidation (Refer to line 423-431) and that this could explain the increase in performance in part.
Query 12: Line 428 and 432, please check this word. “Vldcad”
Reply: We have corrected this typo error.
Query 13: Line 452, “OMM and IMM membrane”, remove the membrane
Reply: This issue was corrected in the MS.

Reviewer 2 Report
Monsalves-Alvarez et al. examined the effect of β-hydroxybutyrate feeding on exercise capacity and muscle mitochondrial function and dynamics, and found that short term feeding of β-hydroxybutyrate increases exercise capacity and mitochondrial protein content, causes mitochondrial morphological changes in mouse skeletal muscle. Though the findings were interesting, but this manuscript contains some serious concerns needed to be modified.
Major
1. Some figures legends says, “… blots cropped from different parts of the same gel,..” . This is acceptable. But “or from different gels were divided by..” is not acceptable. All who have experience of WB know that it is impossible to compare the bands from different gels. Reviewer strongly recommends loading samples again in the order you represent in figure. Another choice is to submit the whole blots you used in this study to explain.
2. Author did an additional 2-week feeding experiment based on results showed in figure 1. However, there are about 30% reduction in distance, time to exhaustion, and work at 2nd and 4th week in control group, which is not observed in Figure 3A. In addition, time to exhaustion longer about 20 min in Figure 3 than in Figure 1.These inconsistence makes the reliability of protocol questionable. Need explanation.
3. One purpose of this study is to examine the effect of β-hydroxybutyrate feeding on muscle mitochondrial morphology. However, it is not clear why authors examine the mitochondrial morphology changes and related protein content in this manuscript. Is there any evidence showing that morphological change directly affect muscle function?
Minor
1. There are duplications of abbreviation, e.g. OMM, lie 75 and 78.
2. Need to show a representative picture of abnormal cristae in figure 7.
3. Discussion is too long.
Author Response
Manuscript ID: Nutrients-833855-R1
Manuscript title: β-hydroxybutyrate increases exercise capacity associated with changes in mitochondrial function in skeletal muscle
Reviewer ‘comments:
REVIEWER 2
Query Regarding language: Moderate English changes required
Reply: The MS was edited by an English native speaker.
Comments and Suggestions for Authors
Monsalves-Alvarez et al. examined the effect of β-hydroxybutyrate feeding on exercise capacity and
muscle mitochondrial function and dynamics and found that short term feeding of β-
hydroxybutyrate increases exercise capacity and mitochondrial protein content, causes
mitochondrial morphological changes in mouse skeletal muscle. Though the findings were interesting, but this manuscript contains some serious concerns needed to be modified.
Major
Query 1: Some figures legends says, “… blots cropped from different parts of the same gel,..” . This is acceptable. But “or from different gels were divided by..” is not acceptable. All who have experience of WB know that it is impossible to compare the bands from different gels. Reviewer strongly recommends loading samples again in the order you represent in figure. Another choice is
to submit the whole blots you used in this study to explain.
Reply: We understand this concern. We attached a file with the whole (intact) membranes of each experiment. We also changed the legend of the Western blot figures to avoid any misunderstanding by readers and added a clarification int the Methods section (Lines 182-183)
Query 2. Author did an additional 2-week feeding experiment based on results showed in figure 1. However, there are about 30% reduction in distance, time to exhaustion, and work at 2nd and 4th week in control group, which is not observed in Figure 3A. In addition, time to exhaustion longer about 20 min in Figure 3 than in Figure 1. These inconsistence makes the reliability of protocol questionable. Need explanation.
Reply: We appreciate your comment to improve our MS. We detected a mistake in our previous analysis in Fig 3. To increase the statistical power of our measurements we added a new set of mice (n=3 per period of treatment) to our original experiments. We increased the number of mice up to 14 in some periods of treatment. Despite this modification, the initial results remain significant in the same period of b-HB treatment (an increase of endurance capacity in b-HB treated mice after two weeks). We also eliminated the original Figure 3. Furthermore, we believe this new statistical
analysis supports the conclusion that the supplementation intake is responsible for the increase of
performance at two weeks.
Query 3: One purpose of this study is to examine the effect of β-hydroxybutyrate feeding on muscle mitochondrial morphology. However, it is not clear why authors examine the mitochondrial morphology changes and related protein content in this manuscript. Is there any evidence showing that morphological change directly affect muscle function?
Reply: Thanks for the valuable comment. Our rationale to relate mitochondrial morphology and protein contents relies on the fact that mitochondrial morphology is highly dependent on nutrient availability such as caloric excess or deprivation (DOI:10.1016/j.cmet.2013.03.002). This issue was unknown in the presence of b-HB.
As the reviewer points out, morphological changes have shown to alter muscle function and exercise capacity. Caffin et al. showed that Opa1 haploinsufficient mice were able to increase their
endurance capacity, despite having an altered mitochondrial ultrastructure morphology (disorganized cristae and modified size) in gastrocnemius muscle, due to a higher fatty acid
oxidation capacity (DOI: 10.1113/jphysiol.2013.263079). These observations are reminiscent of the results we show in our MS. Also, Mammucari et al. (2015) showed that alteration of the mitochondrial calcium uniporter leads to changes in mitochondrial morphology in the EDL muscle
(of glycolytic metabolism as TA) and modification of atrophy markers in SkM. More recently, one of the authors of this article (Sepulveda MC) has shown that mitochondrial size and their
communication with other organelles such as endoplasmic reticulum is highly associated with the lipid oxidative capacity in human SkM (DOI:10.1152/ajpendo.00025.2020). All these observations point to the tight connection between mitochondrial morphology and function in SkM (refer to line
432-436)
Minor
Query 4: There are duplications of abbreviation, e.g. OMM, lie 75 and 78.
Reply: Abbreviation eliminated as suggested.
Query 5: Need to show a representative picture of abnormal cristae in figure 7.
Reply: We marked a “normal” and “abnormal” mitochondrion on the picture and described on the legend (Fig.6).
Query 6: Discussion is too long.
Reply: We reduced the length of the discussion.

Round 2
Reviewer 1 Report
On line 34-36, the authors concluded that “short, but not long-term β-hydroxybutyrate supplementation increases exercise capacity associate to modifications in mitochondrial morphology and function in mouse skeletal muscle.”, however, authors argued that “Here we showed that β-HB supplementation can induce an improvement in exercise capacity” on line 347, and “Still, our study supports the idea that chronic b-HB intake increases exercise capacity, in part, by altering the hierarchy of fuel preferences”. Unlike the claims above, the authors claim that “we cannot completely discard than long term β-HB salts ingestion could lead to unwanted side effects on fatty acid metabolism”, on line 412, these inconsistent argue may confuse the reader.
The data in the study showing that the running capacity was gradually decreased, although b-HB ingestion for 2 weeks resulted in an increase in running endurance. It appears to be related to unwanted side effects on fatty acid metabolism or changed mitochondrial morphology as the authors mentioned on line 411.
I recommend that the authors clarify the discussion.
Author Response
We appreciate this relevant comment on our manuscript. As the reviewer mention, we delete the sentence “Still, our study supports the idea that chronic β-HB intake increases exercise capacity, in part, by altering the hierarchy of fuel preference,” and re-write the lines 411-413 to avoid any confusion by readers.Reviewer 2 Report
no more comment. good work.
Author Response
We appreciate the overall recommendations of the reviewer to improve our manuscript.